

# A global analysis of bird plumage patterns reveals no association between habitat and camouflage

Marius Somveille[1,2], Kate L.A. Marshall[1] and Thanh-Lan Gluckman[1,3,4]

[1] Department of Zoology, University of Cambridge, Cambridge, United Kingdom
[2] The Edward Grey Institute, Department of Zoology, University of Oxford, Oxford, United Kingdom
[3] Department of Animal and Plant Sciences, University of Sheffield, Sheffield, United Kingdom
[4] Center for Interdisciplinary Research in Biology, College de France, Paris, France

## ABSTRACT

Evidence suggests that animal patterns (motifs) function in camouflage. Irregular mottled patterns can facilitate concealment when stationary in cluttered habitats, whereas regular patterns typically prevent capture during movement in open habitats. Bird plumage patterns have predominantly converged on just four types—mottled (irregular), scales, bars and spots (regular)—and habitat could be driving convergent evolution in avian patterning. Based on sensory ecology, we therefore predict that irregular patterns would be associated with visually noisy closed habitats and that regular patterns would be associated with open habitats. Regular patterns have also been shown to function in communication for sexually competing males to stand-out and attract females, so we predict that male breeding plumage patterns evolved in both open and closed habitats. Here, taking phylogenetic relatedness into account, we investigate ecological selection for bird plumage patterns across the class Aves. We surveyed plumage patterns in 80% of all avian species worldwide. Of these, 2,756 bird species have regular and irregular plumage patterns as well as habitat information. In this subset, we tested whether adult breeding/non-breeding plumages in each sex, and juvenile plumages, were associated with the habitat types found within the species' geographical distributions. We found no evidence for an association between habitat and plumage patterns across the world's birds and little phylogenetic signal. We also found that species with regular and irregular plumage patterns were distributed randomly across the world's eco-regions without being affected by habitat type. These results indicate that at the global spatial and taxonomic scale, habitat does not predict convergent evolution in bird plumage patterns, contrary to the camouflage hypothesis.

Corresponding author
Marius Somveille,
marius.somveille@zoo.ox.ac.uk

## INTRODUCTION

Selection for optimal camouflage and sexual signals in the different habitats of animals can drive phenotypic variation (e.g., *Endler, 1978*; *Stuart-Fox & Ord, 2004*; *Hoekstra et al., 2006*; *Rosenblum, 2006*; *Seehausen et al., 2008*; *reviewed in Stevens, 2013*). Animals exhibit a diversity of pigmentation patterns or motifs (i.e., stipples) that are predominantly thought

to function in camouflage, such as by generally matching the background or by breaking up (disrupting) the prey outline and creating false edges to prevent detection by predators (*Thayer, 1909*; *Cott, 1940*; *Pough, 1976*; *Stevens & Merilaita, 2009*). Under sensory ecology theory, the likelihood of a predator being able to detect its prey depends on how effectively the visual cue is transmitted to the predator's eye, and on how well the predator's specific visual system can detect/recognise its prey against the background on which it is viewed (e.g., *Endler, 1992*). Past studies have shown that genetic adaptation of camouflage appears to enhance background matching in local environments and increase survival against visual predators (e.g., *Hoekstra, Krenz & Nashman, 2005*; *Vignieri, Larson & Hoekstra, 2010*). Crucially, visual modelling studies have shown that such local adaptation can prevent detection by the actual visual systems of predators, such as hunting birds (e.g., *Stuart-Fox et al., 2004*; *Marshall et al., 2015a*; *Marshall, Philpot & Stevens, 2015b*), thus indicating that camouflage tends to be optimised to local habitats under natural selection.

Across animals, different types of visual patterns made up of different patches of pigmentation to make up an overall pattern or 'motif' have been linked to different functions in camouflage. Specifically, irregular patterns (mottled) tend to function in stationary camouflage while those that regularly repeat a pattern or motif (bars, spots) function in motion camouflage, which are in turn predicted to be more effective in closed and open habitats, respectively (*Marshall & Gluckman, 2015*). Birds are the most well described taxonomic group of animal and can be found in all types of habitats on all major landmasses (*Gluckman & Cardoso, 2010*; *Jetz et al., 2012*), making them particularly well suited for investigating sensory ecology hypotheses. Recent studies on bird colouration across a large number of species and at large spatial scale revealed that islands are associated with reduced signal intensity (*Doutrelant et al., 2016*) and that colour heterogeneity decreases with body size (*Galvan et al., 2013*), but also showed distinct effects of habitat, life history and sexual selection on the evolution of colouration that lacks patterning (an absence of patterns or uniform colouration), such as bright red carotenoid and dull brown melanins (*Dale et al., 2015*; *Dunn, Armenta & Whittingham, 2015*). However, no study has yet explicitly examined the role of habitat and camouflage on avian feather patterns type at the global scale (*but see Riegner, 2008*; *Gluckman & Cardoso, 2010*).

In spite of the spectacular diversity in phenotypes across the class Aves, bird plumage patterns have predominantly converged on just four types: mottled, scales, bars and spots (Fig. 1; *Gluckman, 2014*; *Marshall & Gluckman, 2015*; *Gluckman & Mundy, 2016*). Most studies of camouflage focus on single or few species (*Marshall & Gluckman, 2015*) that may not be representative of broad scale selection pressures. Here, we investigated whether sensory ecology theory could explain convergent evolution in the camouflage plumage patterns of birds at the global spatial and taxonomic scale, i.e., with a global spatial extent, using large spatial units, and data on most land bird species.

It is thought that irregular patterns function in the camouflage of stationary animals through background matching and disruptive camouflage, in order to evade detection by predators (*Thayer, 1909*; *Cott, 1940*; *Endler, 1978*; *Bradbury & Vehrencamp, 1998*). Accordingly, static irregular camouflage patterns are harder to detect when viewed against more cluttered backgrounds containing distractors (e.g., *Dimitrova & Merilaita, 2009*).
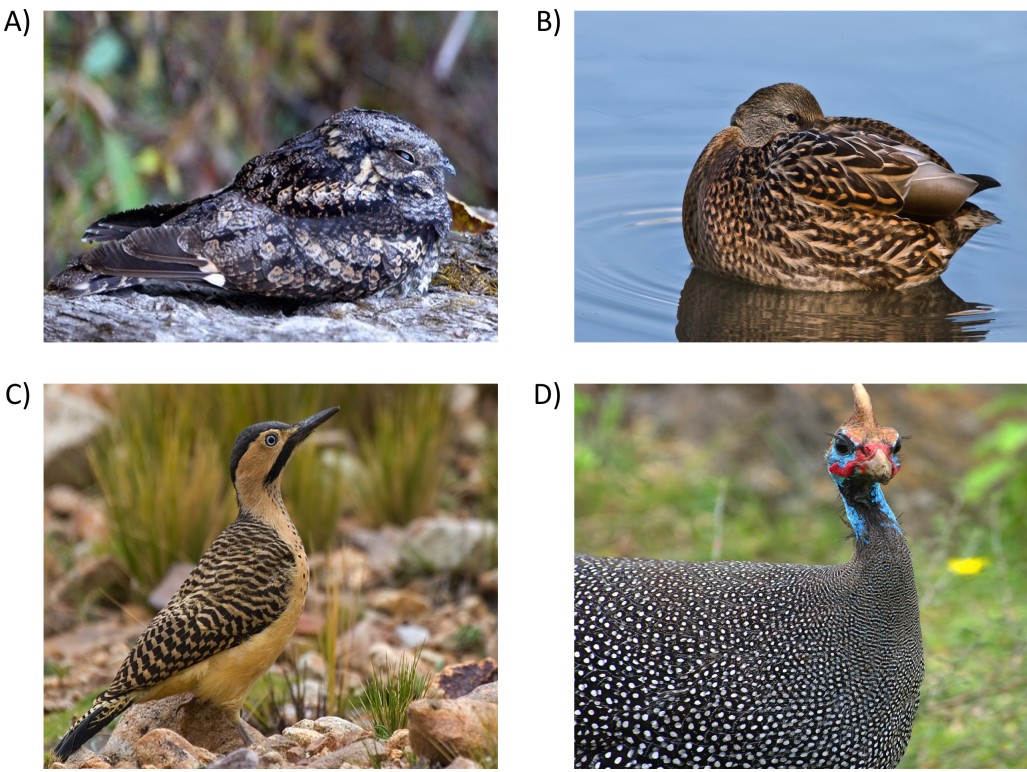

**Figure 1** **The predominant types of plumage patterns in birds.** (A) Irregular mottled plumage consists of feathers that are heterogeneously pigmented (Jungle Nightjar–*Caprimulgus indicus indicus*). Regular plumage patterns can be comprised of (B) scales where pigmentation follows the edge of the vane (Mallard–*Anas platyrhynchos*), (C) bars that are made of alternating dark and light pigmentation transversal to the feathers axis (Andean flicker–*Colaptes rupicola*), and (D) spots where one or more spots pigment each feather (Guineafowl–*Numida meleagris*). Photos: Grey Nightjar.jpg by Koshy Koshy (retrieved from https://commons.wikimedia.org/wiki/File:Grey_Nightjar.jpg CC BY 2.0 license); Female Mallard Duck Rest.jpg by Alain Carpentier (retrieved from https://commons.wikimedia.org/wiki/File:Female_Mallard_Duck_Rest.jpg under a CC-BY-SA 3.0 license); Colaptes rupicola 20070123.jpg by Adam Kumiszcza (retrieved from https://en.wikipedia.org/wiki/Andean_flicker#/media/File:Colaptes_rupicola_20070123.jpg under a CC-BY-SA 3.0 license) and Pintade de Numidie.jpg by JP Hamon (retrieved from https://commons.wikimedia.org/wiki/File:Pintade_de_Numidie.jpg under a CC BY-SA 3.0 license), respectively.

Therefore, irregular patterns should be favoured in visually complex closed environments that provide backgrounds where a prey animal can easily blend in. Irregular mottled patterns are common in avian species, and are more frequently found in females and juveniles in comparison to regular barred plumage patterns (*Gluckman & Cardoso, 2010*).

Current evidence suggests that regular patterns, such as bars and spots, typically facilitate camouflage during movement and therefore function as a secondary defence during the escape of prey by exploiting specific features of receiver visual acuity. This type of camouflage would include motion dazzle, whereby highly contrasting patterns prevent predators from estimating the speed and direction of moving prey (e.g., *Scott-Samuel et al., 2011*; *Stevens, Yule & Ruxton, 2008*; *Stevens et al., 2011*; *Von Helversen, Schooler & Czienskowski, 2013*; *How & Zanker, 2014*; *Hughes, Troscianko & Stevens, 2014*) as well as flicker-fusion camouflage, in which the patterns of a moving animal become blurred

to the predator's vision so that it appears to match the background against which it is moving (e.g., *Brodie, 1989*; *Brodie, 1992*; *Brodie, 1993*; *Lindell & Forsman, 1996*). Given that escape by flight requires space and that motion camouflage is dependent on effective transmission to the predator's eye, regular patterns in birds should be more common in open habitats where patterns are easily visible and escape by flight is unencumbered. Additionally, regular patterns may have a dual function in both camouflage and visual communication with conspecifics (*Marshall, 2000*; *Gluckman & Cardoso, 2010*) and, due to multiple functions, may evolve independently of habitat in adult males (*Bradbury & Vehrencamp, 1998*; *Endler, 1992*; *Kenward et al., 2004*; *Leal & Fleishman, 2004*; *Seehausen et al., 2008*; *McLean, Moussalli & Stuart-Fox, 2014*).

Across the world, habitats differ in their structural composition, e.g., forested habitats are cluttered and visually noisy, while desert habitats are not. The dorsal plumage patterns of birds are likely to function in camouflage in adults of both sexes, as well as juveniles, whereas a communication function is likely to evolve on the ventral surface of sexually competing males (*Stuart-Fox & Ord, 2004*; *Gluckman & Cardoso, 2010*; *Garcia, Rohr & Dyer, 2013*).

In this study, we investigated whether habitat selects for convergent evolution in the plumage patterns of terrestrial birds at the global scale, using the geographical distribution of all known avian species (excluding sea birds). Firstly, we looked for an association between plumage patterns type and habitat type across avian species; and secondly, we investigated whether patterned species were distributed at random across the world. We expected to find a significant association between closed and open habitats with irregular and regular patterns, respectively, in the plumage of adults as well as juveniles. In the sexually selected ventral plumage of breeding males, we expected that plumage pattern evolution would be independent of habitat due to its function in communication. In addition, we expected the distribution of irregularly and regularly patterned species across geographical space to differ from random and be affected by habitat.

## MATERIAL AND METHODS

### Data collection

We focused on the most prominent aspect of patterning: the four different types of spatial arrangement of pigmentation (Fig. 1). To collect plumage pattern information we referred to field guides covering all major landmasses: North and Central America (*Sibley, 2000*; *Van Perlo, 2006*), South America and Antarctica (*De La Pena & Rumboll, 1998*; *Restall, Rodner & Lentino, 2007*), sub-Saharan Africa and Madagascar (*Langrand, 1990*; *Sinclair & Ryan, 2003*), Europe, North Africa and Central Asia (*Heinzel, Fitter & Parslow, 1995*; *Grimmett, Inskipp & Inskipp, 1999*; *Arlott, 2007*; *Arlott, 2009*; *Brazil, 2009*), East and South-East Asia (*MacKinnon & Phillips, 1993*; *Coates & Bishop, 1997*; *Robson, 2005*), and Oceania (*Beehler, Pratt & Zimmerman, 1986*; *Pratt, Bruner & Berrett, 1987*; *Simpson & Day, 2004*; *Robertson & Heather, 2005*). Where multiple subspecies were present, we collected information on the nominate subspecies. Overall, we examined 8006 avian species (80% of class Aves) to categorize their patterning. Juvenile plumages are less frequently drawn in field guides so we were only able to categorise juvenile plumage information in 2,603 species (26% of class
Aves). Species that were unpatterned and/or without habitat information were excluded from further statistical analyses, giving an overall sample size of 2,756 avian species (34% of the species considered), providing information for 2,593 adult and 2,104 juvenile avian species (see the list of species in Appendix S1).

We scored the plumage of both sexes as well as juveniles of each species for: an absence of patterns, mottled, scaled, barred and spotted patterns, both on the ventral and dorsal surface, separately (Fig. 1). Irregular mottled patterns were defined as pigmentation that did not have a clearly discernible motif or the putative motif had irregular borders, as frequently found on the dorsum of sparrows (in contrast with the stripes of species from the genus *Gavia* that have clear and neat boundaries in the stripes on the neck). Regular patterns had a motif with regular borders and were repeated across a patch of patterning (Fig. 1). Some species have multiple patterns on either the ventral or dorsal surface (e.g., the male zebra finch, *Taeniopygia guttata*, has barred patterns on the breast and spotted flanks): on the ventral surface, 62 and 64 species for males during the non-breeding and breeding seasons respectively, 70 and 72 species for females during the non-breeding and breeding seasons respectively, and 63 species for juveniles; and on the dorsal surface, 229 and 222 species for males during the non-breeding and breeding seasons respectively, 231 and 233 species for females during the non-breeding and breeding seasons respectively, and 102 species for juveniles. In these types of species, all of the different types of plumage patterns on each surface were scored and each pattern type was analysed separately. Where species exhibited variable patterns between moults, we collected the breeding and non-breeding plumage given that there may be variation in selection pressure on the different types of patterns exhibited. Squares, triangles and stripes also occur within the plumage of birds, but are comparatively rare (e.g., out of these three rare patterns stripes are the most common: 43 species). Stripes are comprised of regular longitudinal lines (unlike bars, which are pigmented transversal to the feathers axis). These rare types of patterns were excluded from the analysis due to low statistical power.

We examined all of the combinations of gender/age (male/female/juvenile), season (breeding/non-breeding), and body part (ventral/dorsal), individually. Herein we refer to each as *biological combinations*.

## Species geographical distributions

Global geographical distributions of all avian species were obtained from *Birdlife International & NatureServe (2012)* for all species in the plumage dataset and treated as described in *Somveille et al. (2013)*. Briefly, the polygons in the dataset represent the global distribution of species and we included only the polygons for which species presence was coded as extant or probably extant and the species origin was coded as native or reintroduced, and we excluded the polygons corresponding to species in passage (i.e., during migration). In addition, breeding distributions (defined as polygons corresponding to the areas where they are present only during the breeding season) were analysed separately from non-breeding distributions (defined as polygons where they are present only during the non-breeding season). We focused solely on land birds, as the geographical

distribution of marine species is not well known and represents less than 3% of avian species.

We statistically examined our data for an association between habitat and plumage patterns for each *biological combination*. For adult breeding and non-breeding plumages, we used the breeding and non-breeding geographical distributions of bird species, respectively. For juvenile plumages, we used the breeding geographical distributions as this life stage occurs predominantly during the adult breeding season. Note that for resident species (i.e., non-migrants; 85% of avian species), the breeding and non-breeding geographical distributions are the same. Irregular patterning is found in 24% of the 8,006 avian species analysed (1,887 species) whereas regular patterning is found in 23% (1,858 species). Of the regular patterns (scaled, barred and spotted), scaled and spotted patterns are prevalent in 25% (470 species) and 30% (549 species) of avian species (across *biological combinations*) respectively, whereas barring is observed in 66% (1,220 species) of avian species (across *biological combinations*)–percentages are relative to the number of species with regular plumage patterns in at least one *biological combination*.

To avoid an underestimation or overestimation of ecological signal for regular patterns, we categorized regular patterns in two ways for our analyses: (1) all regular patterns grouped together as a single category, and (2) just barred plumage patterns due to these being the most frequent type of regular pattern occurring in bird plumage patterns (*Gluckman & Cardoso, 2009*; *Gluckman & Cardoso, 2010*). The results were qualitatively the same for the analyses of regular patterns together and barred plumage patterns alone, and we present the results for the category of all regular patterns together (a comparison of mottled versus barred patterns for all analyses are available in the Supplemental Information 1).

## Phylogenetic comparative analysis

To examine the habitat in which species live, we analysed the habitat types found within their geographical distributions. For each species with irregular or regular plumage patterns, we quantified how open or closed is the habitat in which they live, a measure herein called *habitat coverage*. Information on habitat was obtained from the global MODIS land cover dataset (*Channan, Collins & Emanuel, 2014*). We coded forests pixels as closed habitat with a value of 1; savannas, grasslands, wetlands, croplands, snow and ice and barren or sparsely vegetated pixels as open habitat with a value of 0; and shrublands, woody savannas and cropland/natural vegetation mosaic pixels as partially closed habitat with a value of 0.5. For each species, we then quantified *habitat coverage* as the average habitat value across the pixels found within the species' geographical distribution. This allowed us to obtain a continuous measure of habitat for each species during both its breeding and non-breeding seasons–which can be very different for migrants, representing ∼16% of avian species (*Somveille et al., 2013*). We then removed species for which no habitat information were available and no plumage patterns were present.

Plumage patterns may be phylogenetically conserved, resulting in similar plumage patterns occurring in closely related species due to phylogenetic inertia (*Felsenstein, 1985*; *Revell, Harmon & Collar, 2008*) rather than caused by habitats selection. In addition, the tendency of species to resemble related species more so than randomly selected species from a tree breaks the assumption of independence of ordinary least square models (*Revell,*
10.7717/peerj.2658
2016
Somveille et al.

_2010_). We therefore examined the association between plumage patterns and habitat by performing comparative analyses that take phylogenetic relatedness into account.

For each _biological combination,_ we computed a phylogenetic logistic regression model (_Ives & Garland Jr, 2010_) with plumage pattern type (regular vs irregular; regular pattern = 0 and irregular pattern = 1) as the dependent variable and _habitat coverage_ as the explanatory variable (R code: phyloglm(plumage pattern type ∼ _habitat coverage_, phy = tree, method = "logistic_IG10")). For each _biological combination_, the analysis was performed only using species with plumage patterns (regular or irregular). As no phylogenetic signal was detected in the models, we also employed standard generalized linear models (GLMs) for comparison (R code: glm(plumage pattern type ∼ _habitat coverage_, family = "binomial")). At large sample sizes, trivially small effects can be detected. Therefore, to examine the statistical significance of the relationship for the GLMs, we evaluated this significance by assessing the goodness-of-fit using the McFadden's pseudo-$R^2$ (_McFadden, 1974_). All statistical analyses were performed in R (_R Development Core Team, 2012_); phylogenetic logistic regressions were performed using the _phylolm_ R package.

Phylogenetic information for bird species was obtained from http://BirdTree.org (_Jetz et al., 2012_). This phylogeny randomly resolves species with missing data by assuming monophyletic genera. To account for phylogenetic uncertainties, we performed our statistical analyses (i.e., phylogenetic logistic regression models) on 100 randomly selected trees and reported the observed variation in the parameter estimates. Branch lengths were computed using the Grafen method (_Grafen, 1989_; R function: compute.brlen). In phylogenetic logistic regressions, _a_ captures the phylogenetic signal in the data and can vary between −4 (no phylogenetic signal) and 4 (strong phylogenetic signal; _Ives & Garland Jr, 2010_).

## Eco-region avian assemblage analysis

To investigate whether species with regular and irregular plumage patterns were distributed randomly across the world, we examined avian assemblages at the scale of the eco-region (_Olson et al., 2001_), using a global map of the world's terrestrial eco-regions made available by _The Nature Conservancy (2009)_ (Fig. S1B). The physical component that defines habitat type are the individual units of land (eco-regions) containing habitat type specific plant communities and species, nested within biomes (_Olson et al., 2001_). We categorized habitat into two main types: closed and open. Closed habitat was defined as broadleaf and coniferous forests as well as Mediterranean forests and the Taiga; and open habitat comprised the various types of grasslands, deserts and tundra as well as the scarce habitats of Inland water, Rock and ice. Mangrove habitat was not considered in our analysis, as it is not clearly composed of solely closed or open habitat.

A bird species was classed as occurring in a given eco-region if its mapped range overlapped with any part of the eco-region. Although this is a coarse species geographical distribution, it represents a good approximation of occurrence given the spatial resolution of the eco-regions (_Hurlbert & Jetz, 2007_). Species richness was measured as the number of species occurring in a given eco-region. After removing eco-regions in which no bird species occur, 791 eco-regions remained, of which 293 (37%) were composed of open habitat and 498 (63%) of closed habitat.

To examine whether the composition of plumage patterns in avian assemblages across the world's eco-regions was random or affected by habitat type, we quantified the ratio of regular to irregular patterns in each eco-region and compared it to a null expectation based on random distribution of the species. The ratio was calculated as the number of species with regular patterns divided by the number of species with irregular patterns (herein referred to as *ratio regular-irregular*). For the analysis comparing barred and mottled patterns, the ratio was calculated as the number of barred species divided by the number of mottled species (results are presented in Supplemental Information 1). To avoid a skew in the distribution, we log transformed this ratio for all of the analyses. Eco-regions without any species with an irregular plumage pattern (2.6% of the eco-regions on average across the *biological combinations*) were excluded from this analysis because the ratio could not be calculated.

The null distribution of plumage pattern ratios across the world's eco-regions was generated by drawing species without replacement 100 times from the hemispherical species pool (i.e., treating Western and Eastern Hemispheres separately). In each eco-region, we drew a number of species corresponding to the observed species richness of the local assemblage, and identified eco-regions with significantly high or low values to the null expectation.

To take spatial autocorrelation into account –whereby species occurring in the local pool of species (i.e., in the neighbouring eco-regions) are more likely to occur in the focal eco-region –we weighted the probability of sampling species based on their degree of occurrence in the local pool. In each avian assemblage (i.e., eco-region), for each species in the hemispherical pool, we determined the proportion of neighbouring eco-regions in which it occurs. Then, for each discrete proportion (i.e., because each eco-region has a discrete number of direct neighbours), we calculated the proportion of species occurring in the focal eco-region. For example, for a given focal eco-region, among all of the species occurring in 50% of the adjoining neighbours, 30% of these species also occur in the focal eco-region. We then fitted a non-linear model (using the nls function in R) to estimate the parameters of this relationship across all the eco-regions (treating the western and eastern hemispheres separately). The fitted curve was then used to determine the probability of a species to occur in a focal eco-region given its occurrence in the neighbouring eco-regions (i.e., the local pool).

We examined the species of the Western hemisphere (the Americas; defined as longitude $<-30°W$) separately from the Eastern hemisphere (the Old World; defined as longitude $>-30°W$) as they have distinctive avian species pools. For example, the total ratio *regular–irregular* (as well as the total *ratio barred–mottled*) is different between these two hemispheres (see Table S2 and Fig. S3). The global species pool from which species were drawn was therefore separated into two species pools representing each hemisphere.

Following the camouflage hypothesis, we expect the ratio *regular–irregular* to be higher in open habitats than the expected null values (e.g., regular patterns dominate), and to be smaller than the null expectation in closed habitats (e.g., irregular patterns dominate). To test this hypothesis, we computed a p-value by comparing the observed value to the distribution of null values. If the eco-region was composed of open habitat, we used

1 minus the cumulative probability because we expect the ratio *regular-irregular* to be higher than the null expectation. To account for multiple testing (i.e., one test for each eco-region) we controlled the false discovery rate using the method developed by *Benjamini & Hochberg (1995)*.

# RESULTS

All types of plumage patterns are found in all biological combinations of females, males and juveniles, for non-breeding and breeding plumage on both the ventral and dorsal surface (Table S1). Mottled plumage is the most prevalent plumage pattern and is found on the dorsal surface more frequently than on the ventral surface in all sex/age classes, and is least frequent in males. Barred patterns are also common and are frequently found on the ventral surface of breeding adult males, but also in juveniles. Scaled and spotted patterns are less prevalent and are frequently biased towards the dorsal surface of adults and the ventral surface of juveniles. However, spotted patterns are also frequently found on the ventral surface of breeding males and females.

## Phylogenetic comparative analysis

Figure 2 shows very little difference in habitat coverage between species with regular versus irregular plumage patterns for all biological combinations, which was confirmed by the statistical models (Table 1). Phylogenetic uncertainty due to randomly resolving species with missing data by assuming monophyletic genera did not meaningfully affect the results (Table 1). For all biological combinations of plumage patterns, *a* was very nearly $-4$ indicating that almost no phylogenetic signal could be detected in the data for plumage patterns. Due to the lack of phylogenetic signal, estimates of intercepts and slopes were almost identical between phylogenetic logistic regressions and GLMs (Table 1). Although most slopes were significantly different from the null hypothesis for both phylogenetic logistic regressions and GLMs, the association between plumage pattern type and *habitat coverage* using GLMs yielded extremely low $R^2$ values (0.001–0.039) with a median $R^2$ of 0.0065 (Table 1). In addition to the low explanatory power, all the models showed a negative relationship between plumage patterns and *habitat coverage*, indicating that regular patterns are more associated with closed environments relatively to irregular patterns that are more associated with open environments, contrary to the camouflage hypothesis.

## Eco-region avian assemblage analysis

Across the world's eco-regions, and age, sex and breeding combinations, avian assemblages had similar proportions of species with regular and irregular plumage patterns, regardless of habitat type (Fig. 3; Fig. S2). In addition, all eco-regions contained qualitatively the same ratio of species with and without plumage patterns for each biological combination, regardless of habitat type (Fig. 3; Fig. S3).

The observed ratio *regular–irregular* in the eco-regions of both open and closed habitats rarely differed from the null expectation (Table 2; similar results were obtained for the ratio *barred-mottled*; Table S4). Less than 1% of eco-regions with species of birds with patterns had a ratio *regular–irregular* significantly lower than the null expectation for any *biological combination* (Table 2), and most of these eco-regions were located on islands.

 9/22

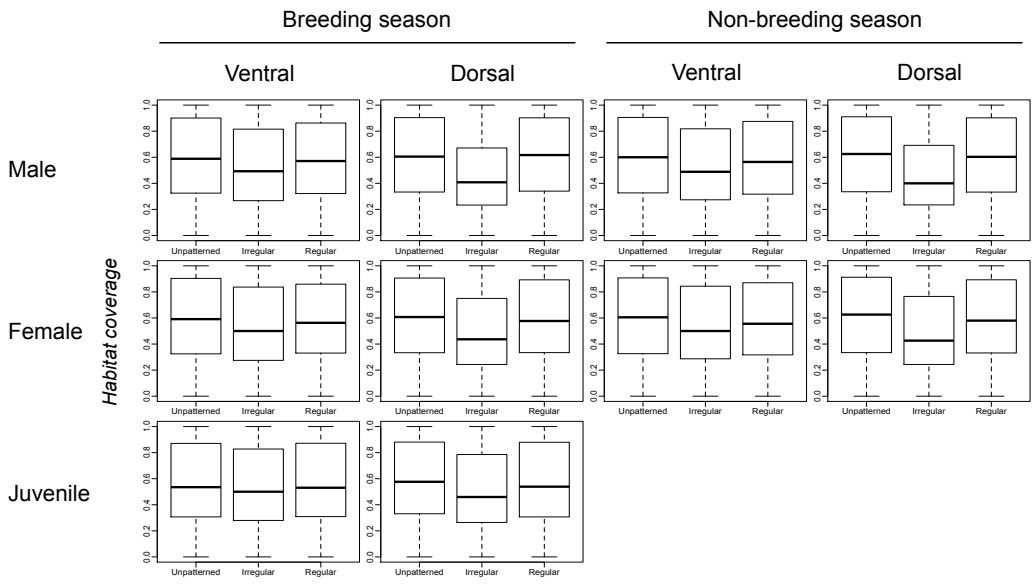

**Figure 2** Comparison of *habitat coverage* values for species without plumage patterns, species with irregular plumage patterns and species with regular plumage patterns across the class Aves. The comparison was plotted for all *biological combinations*.

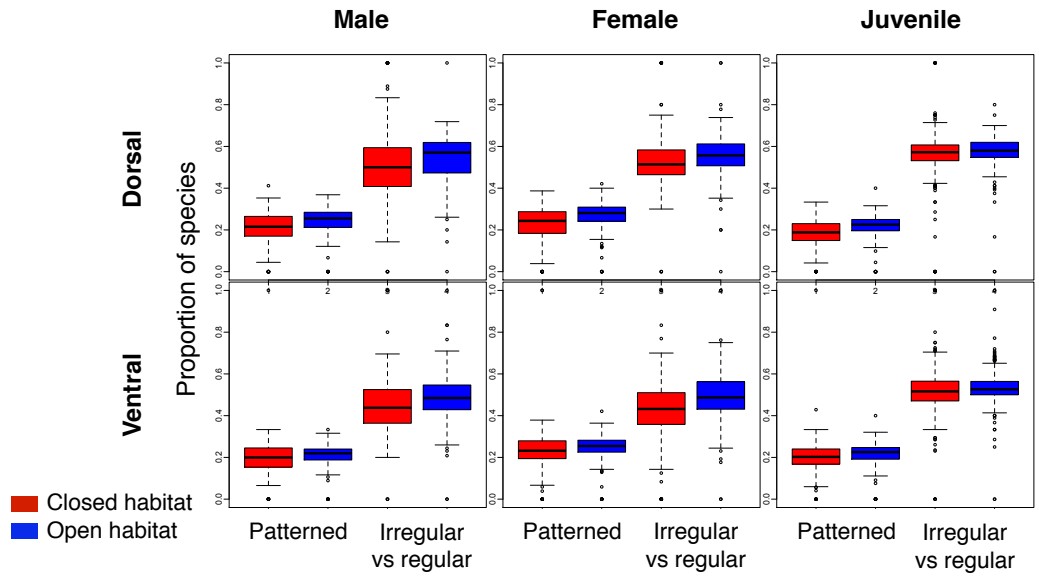

**Figure 3** Comparison of the proportion of species with plumage patterns versus without, and with irregular versus regular patterns in breeding males and females, as well as juveniles, over the dorsal and ventral surface of land birds. The *Patterned* boxplots correspond to the proportion of patterned species in eco-regions' avian species assemblages. The *Irregular* vs *regular* boxplots correspond to the proportion of irregular versus regular patterns in eco-regions' avian species assemblages for the indicated age/sex class. The boxplots in red correspond to closed habitat and the blue boxplots correspond to open habitat.

**Table 1** **Relationship between plumage pattern type (regular versus irregular) and habitat (using our *habitat coverage* measure) across the class Aves using Phylogenetic Logistic Regressions (PhyLoRegs) and Generalized Linear Models (GLMs).** We present the estimate of the intersect as well as the slope and its associated *p*-value for both PhyLoRegs and GLMs. For PhyLoRegs, values correspond to the mean of 100 runs using randomly sampled phylogenetic trees from *Jetz et al. (2012)*, and values in brackets correspond to the standard deviation. Positive slopes indicate that regular patterns are more associated with open habitat while irregular patterns are more associated with closed habitat, and negative slopes indicate the opposite. *a* is the phylogenetic correlation parameter calculated from the PhyLoRegs. $R^2$ values were computed for the GLMs and correspond to the McFadden's pseudo-$R^2$.

| Biological combination | | | | PhyLoRegs results | | | | GLMs results | | | |
| --- | --- | --- | --- | --- | --- | --- | --- | --- | --- | --- | --- |
| Sex | Body part | Season | Raw number of species | Intercept | Slope | P-value | a | Intercept | Slope | P-value | McFadden's pseudo-r2 |
| Female | Ventral | NB | 1,655 | 0.058 ($<10^{-4}$) | −0.298 ($<10^{-4}$) | 0.001 ($<10^{-4}$) | −3.993 (0.016) | 0.058 | −0.298 | 0.067 | 0.001 |
| | | BR | 1,658 | 0.102 ($<10^{-4}$) | −0.375 ($<10^{-4}$) | <0.001 ($<10^{-4}$) | −3.984 (0.026) | 0.101 | −0.375 | 0.022 | 0.002 |
| | Dorsal | NB | 1,643 | 0.769 ($<10^{-4}$) | −1.127 ($<10^{-4}$) | <0.001 ($<10^{-4}$) | −3.986 (0.018) | 0.769 | −1.127 | <0.001 | 0.021 |
| | | BR | 1,648 | 0.786 ($<10^{-4}$) | −1.163 ($<10^{-4}$) | <0.001 ($<10^{-4}$) | −3.983 (0.026) | 0.786 | −1.163 | <0.001 | 0.022 |
| Male | Ventral | NB | 1,336 | 0.171 ($<10^{-4}$) | −0.532 ($<10^{-4}$) | <0.001 ($<10^{-4}$) | −3.983 (0.022) | 0.171 | −0.532 | 0.003 | 0.005 |
| | | BR | 1,334 | 0.147 ($<10^{-4}$) | −0.575 ($<10^{-4}$) | <0.001 ($<10^{-4}$) | −3.973 (0.032) | 0.147 | −0.575 | 0.002 | 0.005 |
| | Dorsal | NB | 1,502 | 0.843 ($<10^{-4}$) | −1.457 ($<10^{-4}$) | <0.001 ($<10^{-4}$) | −3.993 (0.016) | 0.843 | −1.457 | <0.001 | 0.036 |
| | | BR | 1,478 | 0.901 ($<10^{-4}$) | −1.553 ($<10^{-4}$) | <0.001 ($<10^{-4}$) | −3.989 (0.023) | 0.901 | −1.553 | <0.001 | 0.039 |
| Juvenile | Ventral | | 1,443 | 0.384 ($<10^{-4}$) | −0.347 ($<10^{-4}$) | <0.001 ($<10^{-4}$) | −3.981 (0.029) | 0.384 | −0.347 | 0.047 | 0.002 |
| | Dorsal | | 1,394 | 0.700 ($<10^{-4}$) | −0.676 ($<10^{-4}$) | <0.001 ($<10^{-4}$) | −3.985 (0.016) | 0.7 | −0.676 | <0.001 | 0.008 |

**Notes.**

Season

NB, Non-breeding; BR, Breeding.

## DISCUSSION

All types of plumage patterns have evolved in juveniles, females and males in non-breeding and breeding plumage, on both the ventral and dorsal surface (Table S1), and species with regular plumage patterns did not differ on average in the type of habitat in which they occur from species with irregular plumage patterns (Table 1; Fig. 2). In fact, contrary to prevailing sensory ecology based hypotheses of the function of patterns in camouflage, we did not find convincing evidence of an association between habitat type and bird plumage patterns at the global spatial and taxonomic scale (Table 1; Table S2). In addition, we found that species with regular and irregular plumage patterns are distributed at random across the world's eco-regions with respect to one another without any significant effect of the eco-regions' habitat type.

Irregular patterns, which tend to function in stationary camouflage, are unexpectedly found in similar proportion in both closed as well as open habitats that have uniform

**Table 2  Assemblage-level test of an association with habitat type for the ratio *regular-irregular*.** For each *biological combination*, we present the total number of eco-regions that have avian species with plumage patterns, and the number, proportion and habitat type of eco-regions that have an observed ratio *regular-irregular* significantly different from the null expectation.

| Biological combination | | | Number of eco-regions | Significant eco-regions | | |
|---|---|---|---|---|---|---|
| Sex | Body part | Season | | Number | Proportion | Name |
| Female | Ventral | NB | 766 | 2 | 0.003 | Antipodes Subantarctic Islands Tundra (Tundra–Australasia), Pantanal (FGS–Neotropics) |
| | | BR | 759 | 0 | 0 | – |
| | Dorsal | NB | 769 | 0 | 0 | – |
| | | BR | 759 | 1 | 0.001 | New Caledonia Rain Forests (TMBF–Australasia) |
| Male | Ventral | NB | 773 | 0 | 0 | – |
| | | BR | 767 | 2 | 0.003 | Victoria Basin Forest-Savanna Mosaic (TMBF–Afrotropics), Kinabalu Montane Alpine Meadows (MGS–Indo-Malay) |
| | Dorsal | NB | 773 | 2 | 0.003 | Banda Sea Islands Moist Deciduous Forests (TMBF–Australasia), New Britain-New Ireland Montane Rain Forests (TMBF–Australasia) |
| | | BR | 767 | 6 | 0.008 | Admiralty Islands Lowland Rain Forests (TMBF–Australasia), Banda Sea Islands Moist Deciduous Forests (TMBF–Australasia), New Caledonia Dry Forests (TDBF–Australasia), Kinabalu Montane Alpine Meadows (MGS–Indo-Malay), Fiji Tropical Moist Forests (TMBF–Oceania), Bohai Sea Saline Meadow (FGS–Palearctic) |
| Juvenile | Ventral | | 763 | 1 | 0.001 | Antipodes Subantarctic Islands Tundra (Tundra–Australasia) |
| | Dorsal | | 761 | 2 | 0.001 | Zambezian Halophytics (FGS–Afrotropics), Kinabalu Montane Alpine Meadows (MGS–Indo-Malay) |

**Notes.**

Habitat type

TMBF, Tropical and Subtropical Moist Broadleaf Forests;  TDBF,  Tropical and Subtropical Dry Broadleaf Forests;  MGS,  Montane Grasslands and Shrublands;  MFWS, Mediterranean Forests, Woodlands and Scrub;  FGS,  Flooded Grasslands and Savannas;  DXS,  Deserts and Xeric Shrublands.

Season

NB, Non-breeding;  BR,  Breeding.

backgrounds (e.g., deserts) where irregular patterns would probably be easier to detect (e.g., *Bradbury & Vehrencamp, 1998*; *Dimitrova & Merilaita, 2009*; *Hall et al., 2013*). Contrary to our predictions, regular patterns are also found in similar proportion in more visually noisy closed habitats where limited space and obstructed transmission would make it difficult to invoke their function in motion dazzle/flicker-fusion camouflage (e.g., *Brodie, 1989*; *Brodie,*

*1992*; *Brodie, 1993*; *Lindell & Forsman, 1996*; *Stevens, Yule & Ruxton, 2008*; *Scott-Samuel et al., 2011*; *Von Helversen, Schooler & Czienskowski, 2013*; *How & Zanker, 2014*).

The finding that habitat had no effect on the presence of regular patterns on the ventral (breeding) surface of males (in particular barred patterns; *Gluckman & Cardoso, 2010*) would be expected under sexual selection, if regular patterns function solely to maximise conspicuousness to conspecifics. However, visual signals should diverge in biological hotspots due to species recognition/sexual selection (*Bradbury & Vehrencamp, 1998*). Habitat may influence whether the pattern stands out against its background, for example by being conspicuous against a uniform background or opposing the geometric pattern of the background. Indeed, signalling traits can be correlated with ecological characteristics to drive signal divergence, such as in African cichlid fish and *Anolis* lizards (*Leal & Fleishman, 2004*; *Seehausen et al., 2008*) as well as in birds (*Doutrelant et al., 2016*). Other factors, such as whether the pattern is able to convey aspects of individual quality, would influence their evolution independent of the viewing background. This perhaps may explain why barred patterns have repeatedly evolved independently on the ventral surface of males (*Gluckman & Cardoso, 2010*). Conceivably, the same forces of sexual selection/species recognition may have shaped the evolution of the other types of patterns in males, such as spotted patterns (*Roulin et al., 2000*; b; *Petrie & Halliday, 2008*; *Muck & Goymann, 2011*; *Pérez-Rodríguez, Jovani & Mougeot, 2013*).

Several factors could explain the seemingly random distribution of regular and irregular plumage patterns from deserts to tropical forests. First, it could be due to the coarse scale at which we investigated the association between habitat and plumage patterns. Many of the empirical studies that demonstrate a camouflage function of patterns in non-colour changing animals show an association in one or a limited number of species (e.g., *Lovell et al., 2013*; *Kang et al., 2014*; *Marshall, Philpot & Stevens, 2015b.*; *Wilson-Aggarwal et al., 2016*) or were found via predator–prey computer simulations (e.g., *Stevens, Yule & Ruxton, 2008*; *Stevens et al., 2011*; *Scott-Samuel et al., 2011*; *Troscianko et al., 2013*; *How & Zanker, 2014*; *Hughes, Troscianko & Stevens, 2014*; *reviewed in Marshall & Gluckman, 2015*). At the level of microhabitats, some studies demonstrate that individual behaviours may facilitate camouflage, such as a behavioural choice to rest on backgrounds that enhance camouflage (*Tsurui, Honma & Nishida, 2010*; *Lovell et al., 2013*; *Kang et al., 2014*; *Marshall, Philpot & Stevens, 2015b.*; *Troscianko et al., 2016*; *Wilson-Aggarwal et al., 2016*). Under current camouflage theory, microhabitat usage in closed and open habitats should result in irregular patterning being associated with all habitats, as it is typically invoked in static camouflage. However, as regular patterns in camouflage are activated by motion, they would very likely need to be associated with only open habitats. Yet, no association was found at the scale we investigated and both irregular and regular patterns were randomly distributed and showed no difference in their geographic distribution with uniformly coloured avian species and no difference between any biological combination (Figs. 2 and 3; Table 1).

In addition, the visual sensitivities of potential observers can alter the visual affect of patterns in different environments, which we did not consider here. For example, in reef fish, regular patterns become blurred at a distance and match the background to prevent detection by fish predators (*Marshall, 2000*), and certain patterns are imperceptible to some

visual systems and may be highly visible to others in certain environments (e.g., *Cummings, Rosenthal & Ryan, 2003*; *Siebeck et al., 2010*). Thus, avian camouflage patterns may be tuned to deceive predators' visual systems in certain environments by distance-dependent effects and by exhibiting signals that predator visual systems cannot detect (private communication channels). However, to create a distance-dependent effect, regular patterns would require space to blur the pattern and should therefore have been associated with more open habitats, especially on the dorsal surface of females and juveniles, which was not what we found.

Perhaps the use of different strata in the vegetation, e.g., ground-dwelling vs. arboreal, may present different visual microhabitat backgrounds that alter the selection for pattern type. The tops of trees are more open and may allow for more movement required for motion-dazzle/flicker-fusion regular patterns whereas ground-dwelling species may benefit from background matching irregular patterning. Similarly, open habitat such as savannahs and grasslands contain patches of closed environment such as bushes and isolated trees in which static camouflage via irregular plumage patterning might be favoured. Moreover, more closed habitats will be darker due to shade, and predator risk might be lower due to reduced visual perception and increased clutter, so perhaps camouflage in highly cluttered environments is less likely to be favoured by selection. Conversely, in open environments, both regular and irregular patterns are likely to be found for dual functions in camouflage and communication.

Developmental constraint may also explain repeated convergence of the four types of plumage patterns independent of function. For example, white patches of plumage can be attributed to morphogen production (*Price & Pavelka, 1996*) and body size may be negatively related to plumage colour heterogeneity in birds (*Galvan et al., 2013*). *Riegner (2008)* was the first to study the recurrence of plumage pattern elements across the class Aves and emphasised developmental constraint as a mechanism of parallelism rather than convergence. However, patches of pigmentation over the body (e.g., counter shading) were the major focus of this study, and so scales were not included and spots were largely grouped with drab/blended plumage. The results of Riegner's study demonstrated that some plumage patterns shift along a trajectory correlated with body size and that neither habitat nor common descent adequately predicted the evolution of patches of bold pigmentation over the body in aquatic or marine habitats. Also, striped (mottled) patterns appeared to be more frequent in Passerines whereas bars are more prevalent in large bodied species, e.g., Galliformes, raptors, etc. However, *Riegner*'s (*2008*) analysis did not control for phylogeny. Based on a theoretical model of reaction–diffusion based plumage pattern formation (*Prum & Williamson, 2002*), the evolutionary trajectory of plumage in Anseriformes and Galliformes followed the same pathways within and between patches of plumage over the body (*Gluckman & Mundy, 2016*). Given that Anseriformes and Galliformes are diverse in life history attributes, this result warrants further investigation at the macroevolutionary scale. Nevertheless, these studies indicate that there may be developmental constraint in plumage pattern evolution that may have implications for natural selection.

Few patterns may be enough to reduce predation by increasing the number of predator search images, which may also favour polymorphisms within species. For example, there is frequency-dependent selection on alternative morphs of *Tetrix subulata* grasshoppers,

where each morph has varying amounts of mottled patterns in varying colours. However, when all morphs are present in a population, all morphs benefit by providing variation in the search image of predators (*Karpestam, Merilaita & Forsman, 2014*). In the case of birds, perhaps the four different types of patterns, in the context of a community of avian species that may also comprise uniform coloration without patterns, may benefit all member species. Although intraspecific polymorphism is quite different to interspecific variation, intraspecific polymorphism via adaptation to different habitats may be the first step to speciation leading to interspecific variation (*reviewed in Boughman, 2002*; *Stevens, 2013*; *McLean & Fox, 2014*).

In summary, we found no convincing evidence for an association between habitat type and plumage pattern in land birds worldwide, and the distribution of plumage patterns across the world appears to be random rather than affected by habitat type. These results opposes the hypothesis that closed habitat should contain more species with irregular plumage pattern and open habitat more species with regular plumage pattern, as a result of habitat selection for camouflage. This is an intriguing result that implies that plumage pattern evolution does not conform to the prevailing views of selection for camouflage at the global spatial and taxonomic scale.

## ACKNOWLEDGEMENTS

We would like to thank Trevor Price and two anonymous reviewers for insightful comments as well as John A. Endler Nicholas I. Mundy, Fabien Laroche and Andrea Manica for thoughtful feedback as well as Dieter Lukas for technical advice on comparative approaches.

### Funding

This research was funded by an Entente Cordiale Scholarship to M.S., a Biotechnology and Biological Sciences Research Council studentship to K.L.A.M. and the Cambridge Overseas Trust to T-L.G. The funders had no role in study design, data collection and analysis, decision to publish, or preparation of the manuscript.

### Grant Disclosures

The following grant information was disclosed by the authors:
Entente Cordiale Scholarship.
Biotechnology and Biological Sciences Research Council studentship.
Cambridge Overseas Trust.

### Competing Interests

The authors declare there are no competing interests.

### Author Contributions

- Marius Somveille conceived and designed the experiments, performed the experiments, analyzed the data, wrote the paper, prepared figures and/or tables, reviewed drafts of the paper.

- Kate L.A. Marshall conceived and designed the experiments, wrote the paper, reviewed drafts of the paper.
- Thanh-Lan Gluckman conceived and designed the experiments, performed the experiments, wrote the paper, prepared figures and/or tables, reviewed drafts of the paper.

**Data Availability**

We have supplied the raw data used in this analysis for the review process. Please do not publish the raw data alongside this article if accepted as a third-party, who is not a co-author on this paper, partly owns the dataset with T-LG.

**Supplemental Information**

Supplemental information for this article can be found online at http://dx.doi.org/10.7717/peerj.2658#supplemental-information.

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
