# Peer review of "A global analysis of bird plumage patterns reveals no association between habitat and camouflage"

_PeerJ, doi:10.7717/peerj.2658_

## Round 0.1 · original submission · Major Revisions

I don't usually solicit three reviews, but in this case I got lucky and got three positive responses. This obviously means there is interest in your manuscript, and the title alone conveys the importance of the topic. Two of the reviewers suggested only minor comments, but reviewer 3 felt that the paper needed some important revisions, including more attention to the effects of body size and also to the differences between dorsal and ventral coloration. I don't know how challenging it will be to re-analyze the data set or re-cateogorize some of the variables (like habitat, suggested by two reviewers), but I encourage you to explore the reviewers suggestions to the extent that you reasonably can. In the end I may need to send your revision out for a second review, but overall the reviewers conveyed their enthusiasm for your paper.

Reviewer 1 ·

Basic reporting

The article seemingly adheres to PeerJ policies and is generally written in clear English. There are a few points that I found confusing that I have put in the General Comments below.

Introduction and background is adequate, although more background information could be given on animal camouflage and visual systems in general before focusing specifically on birds.

Experimental design

The article identifies a research question which is relevant and of interest: are broad categorizations of plumage patterning associated with habitat types among birds? The methods are generally valid, although I do have a few concerns that are outlined in the general comments below.

Validity of the findings

The authors could provide an appendix that summarizes the MODIS / habitat coverage data used in the logistic regression to improve reproducibility.

Additional comments

Reviewer Summary
This study tests whether habitat coverage is associated with broad categorizations of plumage patterning across birds under a sensory ecology framework. The authors categorized adult plumages of 8006 species of birds (as well as 2603 juveniles) into four plumage types: mottled, scaled, barred, and spotted; mottled was “irregular”, whereas scaled, barred, spotted were treated as “regular”. The authors examined possible association between habitat and plumage types in two ways: (1) phylogenetic logistic regression with irregular or regular plumage as the dependent variable and habitat coverage from MODIS as the predictor variable; and (2) comparing ratios of plumage types among ecoregions categorized as either ‘open’ or closed’. In both tests, the authors do not find evidence for an association between habitat characterizations and plumage types, suggesting that natural selection via camouflage and predator avoidance may not shape interspecific variation in plumage patterning at a macroevolutionary scale.

This study is motivated by an interesting biological question, and the methods are largely valid, although I have some concerns about the way that the Jetz (2012) phylogeny is used, the decision to include both dorsal and ventral patterning, and the exclusion of species that have no patterning (see below for more detail). The writing is generally clear, although there are some minor typos and a few confusing passages throughout that could be improved in future revisions. I think some details need to be expanded upon in the methods to help readers understand what exactly the authors did and why they did it. I hope that the authors find my comments useful.

Reviewer Comments
Title: Consider revising the title. In its current form, the title suggests that the paper finds positive evidence for “Ecological selection for bird plumage patterns worldwide”, while the authors “report no evidence for an association between habitat and plumage patterns across the world’s birds”.

Validity of the Jetz phylogeny: The authors should clarify whether they used the Jetz phylogeny that only includes species for which genetic data exist, or whether they used the Jetz phylogeny that randomly resolved species with missing data by assuming monophyletic genera. If it is the latter, the authors should discuss potential pitfalls of using this phylogeny. It might be advisable to perform analyses on individual phylogenies rather than a consensus tree to account for potentially erroneous relationships in some of the Jetz trees and variation in branch lengths among phylogenies sampled from the posterior. The authors could also restrict their analysis to only those species with genetic data in the Jetz tree to see if this aspect of the phylogeny influences their inferences.

It’s not clear to me why the authors use both tests presented here (i.e., phylogenetic logistic regression with irregular vs regular as response variable and comparison among ecoregions with ratio of irregular vs regular as response) since they are essentially testing the same thing: is there a statistical association between habitat characteristics and plumage patterns? I understand the motivation to look at multiple lines of evidence, but the authors should motivate why including both tests is preferable to either single test to help guide the reader. What does the ratio test among ecoregions provide that is not gained by the logistic regression, which in my opinion, has a more accurate and elegant quantification of habitat variation?

I’m confused by the decision to not include unpatterned species in the study. If an organism has ‘uniform’ plumage on its dorsal side that matches the substrate, is that not considered some form of camouflage or a pattern of sorts? Might uniform coloration be a type of ‘regular’ patterning? Along these lines, it is somewhat misleading to say that the study is based on >8000 species if those that do not have any patterning are not included in the study. In reading further, I realize that these are included in Fig. 3, but the methods are confusing in the way they are currently written with respect to how non-patterened species are considered in this study.

A brief explanation of what motion dazzle and flicker-fusion camouflage is and how it works would be useful for uninitiated readers.

Line 49–51: This passage is confusing as written “pattern camouflage needs to be transmitted to a predator effectively…”. Organisms that use camouflage try not to be detected by predators, whereas this suggests that they are purposefully relaying some kind of information to predators. Upon reading further, I’m guessing that this refers to the ‘motion dazzle’ or flicker-flash signals, but this should be clarified—it is confusing in its current form.

Line 141–142: What aspect of the evolutionary history of barred plumage rationalizes this decision? Unclear as written, even though it didn’t make a difference and the authors decided to lump all regular patterns together.

Line 168–170: I don’t completely understand why the authors chose to run regular GLMs in addition to the phylogenetic logistic regression. The Ives and Garland (2010) method estimates and accounts for phylogenetic signal already, so why are regular GLMs also needed?

Line 267–271: The directionality of the logistic regression (i.e., the fact that a negative slope means what is described in this passage), could also be included in the table legend.

Line 292–293: The construction “at large scale” is used frequently in this study. I find it reads odd and I would consider revising it. Maybe something like “at a broad taxonomic scale” or “macroevolutionary scale”.

Line 322–324: Please provide some citations to studies (or better yet a review) that discusses camouflage at the microhabitat scale.

Table 1: Table legend is misleading in that it suggests that habitat characterization here is binary (i.e., open vs closed), when in fact it is continuous ‘habitat coverage’ based on MODIS data.

·

Basic reporting

all seems fine

Experimental design

OK, one point is highlighted in the author's comments below

Validity of the findings

fine

Additional comments

For Peer J this paper seems fine and a useful contribution. It does appear to me that the four categories considered could all be considered solutions to the same general problem of breaking up an immaculate pattern. The main methodological issue I had was considering habitat in a species range to be the habitat a species occupied. It would have been better to have simply classified species according to their habits, which are crudely available, although I certinaly wouldn’t make this a requirement.
l.78: I am not sure that phylogenetic relatedness is a “cause”, but more the point is that related species share similar habitats, environments?
l.105 species not spp.
l.116 what is a stripe?
l.128 clearly define “biological combination” on its first presentation
l.134 it would be of interest to show how many show both (as in the zebra finch), perhaps in a full table.
l.198 what is not always true?
Price & Pavelk (1996). J. Evol. Biol. 9:451-470 examined the appearance of the crown stripe across the class Aves and used simple reaction diffusion models to show why it may be more often stripe than – say – a cross.

Reviewer 3 ·

Basic reporting

This is a nice comparative study testing for associations between plumage patterns and habitat types in a large number of birds. The main hypothesis is simple (i.e. irregular patterns should have evolved in species occupying closed habitats as they favor stationary camouflage, while regular patterns should have evolved in species occupying open habitats as they favor camouflage during movement), but could help understanding some basic selective pressures shaping the diversity in bird plumage patterns. I have, however, some concerns that should be addressed.

Experimental design

- Lines 148-157. The association between plumage patterns and habitat was tested using habitat coverage as a description index of habitat type. For this, the proportion of different habitat types was quantified within the geographical distribution area of each species. In my opinion, this is a fallacy, as species only occupy certain habitats along their dsitribution ranges. Thus, the authors should categorize the species in relation to the habitat type that they actually occupy, an information that can probably be obtained from the same field guides from which plumage descriptions were taken. I suggest to categorize habitat types in a little more detail, for example differentiating between closed, medium (disperse vegetation) and open habitats. Otherwise categories will largely overlap, which will prevent finding any patterns.

- Also in relation to the previous point above, the results of this study will not be reliable if the analyses are not controlled for species body size. Body size is negatively related to plumage color heterogeneity in birds (Galvan et al. 2013, Acta Ornithologica 48: 65-80), so it should be considered in any comparative study of plumage patterns. Furthermore, in this case any effect of habitat cannot be elucidated without considering that small species are obviously more constrained to exhibit patterns for camouflage than larger species with less predation pressures.

- Line 115. It is stated here that squares, triangles and stripes are comparatively rare in the plumage of birds. I agree that squares and triangles are rare plumage elements, but do not think the same about stripes and find hard to believe that the authors only found 43 or fewer species with stripes in the plumage. At a glance, I can mention several common European birds with stripes, like many herons (e.g. squacco heron, little bittern), raptors (juvenile Northern goshawks, juvenile Montagu's harrier) and many passeriforms (dorsal patterns in some buntings such as the common reed bunting or the Lapland longspur to mention only a few). Therefore I encourage the authors to reconsider their analyses, or to clarify their concept of stripe. In general, I think that the concept of pattern here should be clarified. For example, the great tit has a single ventral stripe. Is this a regular pattern?

- Lines 248-256. I cannot understand why the authors did not conduct statistical analyses to test for differences in the frecuency of pattern types in dorsal and ventral sides. The information in Table S1 should only be analyzed with frequency tests whose results should be reported in this paragraph. This is a great opportunity to determine if regular patterns are more frequent in ventral plumage patches than in dorsal patches if they have a intraspecific signaling function and not only a interspecific signaling function related to camouflage. If there are no differences in the appearance of regular patterns between ventral and dorsal sides, the authors should mention that these results do not support the suggestions of some authors that certain mathematical indexes for regular plumage patterns describe information on individual quality (Perez-Rodriguez et al. 2013), as such lack of differences would suggest that regular patterns are mainly related to camouflage and not to signalling of individual quality.

Other comments:
- Lines 110-112. Were there any species showing different types of pattern on the same surface (ventral or dorsal)? This should be clarified.
- Lines 122-124. Please describe briefly this method here.

Validity of the findings

No Comments.

---

## Round 0.2 · accepted · Accept

Thank you for the revision of your manuscript. You have addressed all the reviewers' comments adequately and provided additional appendices and data to improve transparency.